# Diagnostic Value and Prognostic Significance of Nucleated Red Blood Cells (NRBCs) in Selected Medical Conditions

**DOI:** 10.3390/cells12141817

**Published:** 2023-07-09

**Authors:** Katarzyna Pikora, Anna Krętowska-Grunwald, Maryna Krawczuk-Rybak, Małgorzata Sawicka-Żukowska

**Affiliations:** Department of Pediatric Oncology and Hematology, Medical University of Bialystok, Jerzego Waszyngtona 17, 15-274 Bialystok, Poland; kat.pikora@gmail.com (K.P.); maryna.krawczuk-rybak@umb.edu.pl (M.K.-R.)

**Keywords:** NRBC, nucleated red blood cells, biomarker, neonates, intensive care unit

## Abstract

Nucleated red blood cells (NRBCs) are premature erythrocyte precursors that reside in the bone marrow of humans of all ages as an element of erythropoiesis. They rarely present in healthy adults’ circulatory systems but can be found circulating in fetuses and neonates. An NRBC count is a cost-effective laboratory test that is currently rarely used in everyday clinical practice; it is mostly used in the diagnosis of hematological diseases/disorders relating to erythropoiesis, anemia, or hemolysis. However, according to several studies, it may be used as a biomarker in the diagnosis and clinical outcome prognosis of preterm infants or severely ill adult patients. This would allow for a quick diagnosis of life-threatening conditions and the prediction of a possible change in a patient’s condition, especially in relation to patients in the intensive care unit. In this review, we sought to summarize the possible use of NRBCs as a prognostic marker in various disease entities. Research into the evaluation of the NRBCs in the pediatric population most often concerns neonatal hypoxia, the occurrence and consequences of asphyxia, and overall neonatal mortality. Among adults, NRBCs can be used to predict changes in clinical condition and mortality in critically ill patients, including those with sepsis, trauma, ARDS, acute pancreatitis, or severe cardiovascular disease.

## 1. Introduction

Nucleated red blood cells (NRBCs) are premature erythrocyte precursors rarely present in the circulatory systems of healthy adults [1,2]. They normally reside in the bone marrow of humans of all ages (Figure 1), where the common myeloid progenitor cell differentiates into more developed cells to finally become an erythroblast [3,4,5]. At this stage, the nucleus is expelled, and the cell becomes a reticulocyte, later developing into a mature erythrocyte [6]. NRBCs can be found in circulating in fetuses and seem to disappear in healthy neonates in the first month of life [7,8,9]. An increased NRBC count in neonates or the appearance of NRBCs at a later age are thought to be associated with the release of NRBCs from the bone marrow due to numerous medical conditions, such as blood loss or hypoxia; therefore, NRBCs can also be observed in the peripheral blood of healthy adult blood donors [10,11]. Their presence in the blood can indicate increased red blood cell production outside the bone marrow or a disruption of the blood–bone marrow barrier [8].

The prompt diagnosis of prenatal conditions is crucial for ensuring proper neonatal care. Several studies have identified NRBCs as biomarkers in diagnosing increased erythropoiesis, acute and chronic hypoxia, anemia or hemolysis, and blood loss [12,13,14]. Christensen et al. showed that this easily accessible and low-cost analysis might be additionally useful in the diagnosis of other medical conditions and can be identified as a prognostic marker among severely ill patients. Efforts were also made to establish the most optimal reference values for NRBCs among different age groups [15].

Numerous studies have revealed that NRBCs can be used as an important indicator of the presence and duration of intrauterine hypoxia or the assessment of the severity and early outcome of perinatal asphyxia [16,17]. NRBCs are also a sensitive indicator of mortality among preterm infants and the prognosis of conditions such as retinopathy of prematurity (ROP), bronchopulmonary dysplasia, necrotizing enterocolitis, or sickle cell disease [4,18,19,20,21].

Among adult patients, NRBC count assessment appears to be the most significant predictor of mortality among patients who have suffered trauma and currently suffer from sepsis and other critical conditions [22]. In hematology, it may be used as an additional marker for predicting treatment efficiency among patients with chronic myeloid leukemia (CML) and for forecasting the course of myelodysplastic syndromes (MDS) and their potential risk of transformation into acute myeloid leukemia (AML) [23,24].

Yet, to date, there are very few studies focusing on the integration of NRBC measurement into clinical decision making. In a prospective study by Stachon et al., critically ill patients in an intensive care unit were monitored based on their NRBC levels. This enabled the physicians to select those patients who were at greater risk of a poor outcome and required closer monitoring [25]. Hebbar et al., on the other hand, progressively evaluated the risk of negative neonatal outcome among children of mothers with pre-eclampsia based on their NRBC levels. This led to the selection of babies at a greater risk of neonatal intensive care unit admission [8]. These studies showed great promise with respect to the potential clinical use of NRBCs as a prognostic marker in the selected medical conditions mentioned in this review article. 

Life-threatening conditions require quick and effective measures for diagnosing patient deterioration. Finding biomarkers that aid in the prompter identification of critical situations among both adult and pediatric (especially with respect to neonates) populations is crucial for efficient medical intervention in order to ensure the highest possible chance of a recovery. In this review, we set out to thoroughly analyze the diagnostic value and prognostic significance of NRBCs in selected medical conditions, a subject that, to date, has not been reported to such an extent in the English language literature.

## 2. Materials and Methods

Extensive research was conducted in June 2022 on the PubMed online electronic database using the key phrases “nucleated red blood cells” or “NRBC”. Publication data restrictions were not assigned. Ninety-three articles were found in total, and the further selection of the articles was conducted according to the preferred reporting items for systematic reviews and meta-analyses (PRISMA) guidelines. Additionally, the review was registered with the Open Science Framework Registries (OSF)—https://doi.org/10.17605/OSF.IO/YXQG3. Before the initial selection, 5 articles were noted to contain duplicates and were thus removed. In addition, 4 articles were excluded because they were written in a language other than English. Five articles could not be retrieved. Of the seventy-nine reports that were evaluated for eligibility, fifteen were rejected because they contained information irrelevant to the study presented, and three were rejected because they concerned only technical aspects. One article was excluded because it did not relate to human subjects. A total of sixty articles were finally analyzed in this review. Figure 2 shows a diagram of the selection process of the scientific papers selected for this article. 

## 3. NRBCs as a Biomarker in Neonatology

### 3.1. NRBCs as an Indicator of Hypoxia among Neonates

Neonatal hypoxia and ischemia remain the most common causes of disability and death among neonates and are often associated with persistent motor, sensory, and cognitive disturbances (accounting for 23% of infant mortality worldwide) [27]. Therefore, a quick diagnosis of hypoxia is crucial for the immediate initiation of appropriate medical therapy.

According to several studies, NRBC count at birth may be a biomarker of neonatal hypoxia in terms of both its duration [16] and estimated severity [28]. Scientists have taken advantage of this fact in an attempt to associate NRBC counts with specific causes of hypoxia and predict their impact on further infant development. 

A study by Christensen et al. attempted to correlate the time between the onset of intrauterine fetal hypoxia and NRBC appearance in the blood. This was based on the time required for erythropoietin (EPO) production to begin in response to hypoxia. The study retrospectively analyzed cases of neonates born at 34 weeks of gestation who were given two doses or a dose of darbepoetin after birth. NRBCs first appeared in the peripheral blood between 24 and 36 h after administration. Patients receiving higher doses tended to have a higher peak in their NRBC counts, but the timing of NRBC appearance was not dose-dependent. Since the process of increasing plasma EPO levels takes from 4 to 5 h, the authors concluded that neonates with elevated NRBC counts at birth had an onset of hypoxia at least 28 to 29 h before birth [16,29,30].

NRBC levels among infants may additionally correlate with the need for higher concentrations of oxygen in the breathing mixture and with increased pulmonary vascular resistance among preterm infants treated at birth with a surfactant. It has been shown that infants with elevated NRBC levels require higher concentrations of oxygen than those with normal NRBC levels. NRBCs correlated positively with indices responsible for estimating pulmonary vascular resistance. In view of these results, it was hypothesized that intrauterine hypoxia may play a role in postnatal circulatory adaptation in neonates with respiratory distress syndrome [31].

Elevated NRBC counts among neonates may indicate intrauterine hypoxia and brain damage in the form of hypoxic ischemic encephalopathy (HIE) [32]. Li et al. retrospectively examined the outcomes of dozens of neonates with HIE by evaluating brain-imaging changes viewed via MRI 2 weeks after birth and then through continued observation for 2 years. NRBC levels were examined for all newborns during the first 3 days of life. It was shown that both the absolute number of NRBCs and the ratios per 100 white blood cells (WBC) were significantly higher in children with HIE than in the control group. Additionally, infants with higher numbers had an increased risk of having abnormal MRI findings and worse outcomes at their 2-year follow-up [33]. A study by Walsh et al. sought to evaluate the value of NRBC counts and electroencephalography (EEG) scores among neonates with HIE as a marker of the disease’s severity. NRBC counts and continuous multichannel EEG recordings were recorded during the first 24 h of life, and a neurological evaluation was conducted after 2 years. The median NRBC counts were significantly higher in infants with moderate to severe HIE than those with mild HIE. Additionally, the combined marker from the two tests (NRBC count and EEG) at each time point showed a stronger correlation with HIE severity than EEG alone [34]. 

Other issues raised in scientific reports include the effects of maternal overweight and obesity on fetal status, prognosis, and the association of these factors with NRBCs. Increased rates of Caesarian sections, postpartum hemorrhage, and macrosomia have been observed in mothers with an elevated BMI. A study by Gohir et al. concluded that the placentas of obese mouse mothers were found to have immature blood vessels, tissue hypoxia, and elevated levels of inflammatory markers [35]. Another study comparing the placentas of obese and non-obese mothers demonstrated a positive relationship between maternal obesity and cord blood erythropoietin levels at birth [36]. Additionally, Persson et al. suggested that the risk of severe asphyxia among newborns born at term increases when mothers are overweight and obese, which also indicates that preventing overweight among women of reproductive age is important for improving perinatal health [37]. All these studies indicate that maternal obesity may be associated with chronic fetal hypoxia. Assuming that the NRBC count test can be a useful indicator of neonatal hypoxia, the validity of performing this test can also be considered as a screening test for overweight pregnant women.

Additionally, a significant positive correlation was found between erythropoietin and NRBC levels in the cord blood and maternal BMI, which may indirectly predict poor neonatal outcomes [28]. NRBCs may also be a marker of obstetric and postpartum complications among low-birth-weight infants. After examining the percentage of NRBCs per 100 WBCs in fetuses small for their gestational age and with determined normal umbilical artery Doppler findings, it was shown that the group with a higher percentage of NRBCs had a significantly higher incidence of Caesarian sections, fetal distress, acidosis, and lower terminal birth weight [38]. Another maternal factor in the occurrence of neonatal complications is the presence of diabetes. A study by Namavar Jahromi et al. compared the outcomes of newborns of both healthy and diabetic women requiring insulin. The mean NRBC counts in the newborns of diabetic mothers were statistically significantly higher, as was the arterial blood carbon dioxide partial pressure, while the pH values were lower. The coexistence of these three relationships may indicate chronic intrauterine acidosis among newborns of diabetic mothers [39]. Additionally, the number of NRBCs in newborn populations may be an exponent of hypoxia expressed through birth meconium in the amniotic fluid (MSAF). Elsokkary et al. compared the results of forty women with clear amniotic fluid and forty women with meconium-stained amniotic fluid. The mean number of NRBCs in the cord blood of neonates with MSAF was significantly higher than that in the control group, supporting the hypothesis that the presence of meconium may be associated with chronic fetal hypoxia [40].

### 3.2. NRBCs as a Marker of Infant Mortality

According to several clinical studies, an infant’s NRBC count at birth may be an independent predictor of mortality among preterm and term born infants [41,42,43]. A 2015 study evaluated the NRBC counts in the peripheral blood of several hundred premature infants weighing less than 1500 g analyzed within the first 5 days after birth. The neonates who did not survive had significantly higher NRBC counts between days 2 and 5 compared to the other group. A 0.01/µL increase in the mean NRBC count resulted in a statistically significant increase in the probability of severe disease morbidity and mortality. Therefore, it was hypothesized that the NRBC count measured on these days might be an independent predictor of early infant mortality [18].

In recent years, attempts have also been made to investigate the prognostic value of NRBCs with respect to critically ill children of different ages. For this purpose, clinical presentation, manifested by the need for inotropic positive drugs, dialysis, or mechanical ventilation, and clinical outcome have been compared with hematological parameters and the severity of the underlying disease itself. In the group of studied neonates, patients who died, were provided assisted ventilation, or received inotropic drugs had significantly more NRBCs in their peripheral blood. Among older children, the number of NRBCs was only found to be correlated with the need for renal replacement therapy. Therefore, NRBC count cannot be considered as an overall independent factor for predicting poor outcomes in pediatric intensive care. However, it may have prognostic significance among children in the first month of life. It is presumed that an explanation might lie in the differences in interleukin-6, erythropoietin (EPO), or catecholamine production between the two age groups [41]. 

Another group of infants studied consisted of those requiring extracorporeal membrane oxygenation (ECMO) because of cardiac dysfunction or a cardiotomy. Piggott et al. suggested that a >50% increase in NRBC count after ECMO decannulation correlates with in-hospital mortality and that pre-ECMO NRBCs may be a useful biomarker of mortality dur-ng ECMO therapy and after decannulation [44]. Additionally, Piggott et al. sought to determine whether an elevated NRBC count after cardiac surgery and subsequent hospitalization could be used as a marker to estimate the risk of neonatal mortality after such surgery [45]. Morton et al. examined the correlation between NRBC counts and deaths among patients in intensive care units (ICU). Among all patients admitted to the ICU, the presence of elevated NRBCs was associated with increased mortality [7]. As there seem to be no specific markers of clinical outcome for ICU neonate patients, an elevation in NRBC count should always be considered as a possible poor prognostic factor of a patient’s condition.

### 3.3. NRBCs as a Prognostic and Diagnostic Factor in Perinatal Asphyxia

Asphyxia can severely affect the major organs of neonates and lead to respiratory distress syndrome, disseminated intravascular coagulation, subcutaneous fat necrosis, myocardial ischemia, adrenal hemorrhage, or neurological complications [46]. Asphyxia is one of the major causes of neonatal mortality and chronic neurological disorders among surviving neonates. Recent reports in the literature have shown that the number of NRBCs in the bodies of newborns with asphyxia may be correlated with the disease’s severity and indicate the development of complications and the overall prognosis of children [47,48,49]. Goel et al., upon examining 100 newborns, confirmed the existence of a relationship between the severity of hypoxic ischemic encephalopathy and NRBC count. Additionally, both neonates with lower Apgar scores and those who did not survive had higher NRBC levels in their cord blood. The overall predictive accuracy of the marker was determined to be 96%, indicating that it can be used as a simple indicator to assess the severity and predict the prognosis of neonates before discharge [17]. Boskabadi et al. attempted to determine whether NRBC levels could be a prognostic indicator of later development among infants with developed asphyxia. An NRBC count of more than 11 per 100 WBCs or an absolute NRBC count above 1554 were shown to have high sensitivity and specificity with respect to predicting complications of asphyxia. These results may also demonstrate the potential use of NRBCs as prognostic indicators in the future [50].

Additionally, Rai et al. focused on assessing typical neurological complications among neonates with asphyxia. NRBC counts were significantly higher in newborns developing abnormally, newborns who had seizures within the first 12 h of life, developed stage 3 HIE, and/or required anticonvulsants [47]. The obtained results prove that NRBCs may be a helpful marker for predicting neurological development among newborns with a history of perinatal asphyxia.

## 4. NRBCs in Hematological Conditions

To date, very few reports on the significance of NRBCs in hematological disorders have been published in the English language literature. One such study aimed to investigate the role of NRBCs in predicting remission failure in adult patient populations with chronic myeloid leukemia (CML) treated with Imatinib. The results demonstrated that median NRBC levels decreased during treatment, and NRBCs were nearly almost always absent in the blood of patients with good responses at the time points of 6, 12, and 18 months of therapy. In addition, CML patients with high NRBC levels in the peripheral blood had increased levels of BCR-ABL transcripts. This facilitates the potential use of NRBC as a biomarker of molecular remission failure in imatinib-treated CML patients [23].

On the contrary, a study by Jiang et al. focused on the role of nucleated red blood cells in the development of myelodysplastic syndromes (MDS). It was hypothesized that the impairment of mitochondria and mitophagy may be involved in the pathophysiology of MDS. NIX-induced mitophagy was found to be disturbed in the NRBCs of adult MDS patients. The results showed a negative correlation between the number of mitochondria in the NRBCs and hemoglobin levels among high-risk patients diagnosed with myelodysplastic syndromes [51].

## 5. NRBCs as a Marker of Bad Prognosis and Mortality in Severely Ill Patients

The association between NRBC counts and the clinical prognosis of patients in intensive care units (ICUs) has been studied quite widely, with several attempts made to determine whether NRBCs could constitute a potential biomarker. A study by Moura Monteiro Junior et al. attempted to determine the role of NRBC levels in the prognoses of ICU patients with cardiovascular episodes. NRBCs were detected in the peripheral blood of over half of the patients with higher NRBC levels observed during longer hospital stays, in older patients, in patients with conditions complicated by sepsis, and in those with non-coronary heart disease. The presence of NRBCs was associated with an increased risk of mortality both in the ICU and during subsequent hospitalization in other departments. The predictive value of this marker was assessed to be independent and could be used as an adjunct to the well-validated Acute Physiology and Chronic Health Evaluation II classification (APACHE II) score [52].

Additionally, Narci et al. examined whether NRBCs could be a predictor of all-cause mortality among patients admitted to the emergency department, with trauma patients excluded. Interestingly, NRBC levels were significantly higher in deceased emergency department patients [22].

Notably, Menk et al. presented similar results with regard to patients with acute respiratory distress syndrome (ARDS). The presence of NRBCs in ARDS patients was identified to be an independent risk factor for death, with their incidence in the peripheral blood doubling the risk of ICU related mortality [53].

A study by Xu et al. aimed to develop a predictive model that could determine the outcomes of patients with severe acute pancreatitis based on NRBC counts. Interestingly, the NRBC-positive rate among non-survivors was significantly higher than that among those who survived. The concomitant occurrence of a negative NRBC test result and a Charlson Comorbidity Index (CCI) below seven predicted survival among patients with acute pancreatitis. On the other hand, all the patients who died exhibited a positive NRBC test result and an APACHE II score greater than 30. The results of the study show that NRBC counts combined with CCI, APACHE II, and Ranson scores can be a predictor of 90-day mortality among patients with acute pancreatitis [54].

NRBC counts were also evaluated as an early prognostic marker of death among patients with surgical sepsis. Desai et al. determined that surgical sepsis was more likely to result in death among NRBC-positive patients both in the ICU and other hospital departments. A peak NRBC count of over 500/µL at any time was associated with 50% or higher mortality; additionally, only one out of nine patients survived their hospital stay when the peak NRBC count was over 2000/µL. The difference in mortality was particularly marked among surgical patients with severe sepsis. This study suggests that NRBCs may be a biomarker of survival outcome among surgical sepsis patients [55].

In addition, Shah et al. attempted to determine whether NRBC counts could be a prognostic marker for surgical intensive unit patients’ clinical outcomes. Patents with no NRBCs detected in their peripheral blood were associated with shorter periods of hospital admission. An elevation in NRBC levels was associated with an increase in mortality. Trends in changes in NRBC values showed that a decrease to an undetectable value had a protective effect. Notably, there were no statistically significant differences between the groups of ICU patients with and without trauma [56].

Furthermore, Kuert et al. evaluated the effect of low arterial blood oxygen partial pressure (pO_2_) on NRBC incidence and prognosis among NRBC-positive patients. Patients were considered NRBC-positive when NRBCs were identified at least once during their hospitalization in the intensive care unit. According to this study, NRBC detection in the peripheral blood of intensive care unit patients correlated with significantly higher mortality than among patients with negative NRBC levels. Interestingly, all patients with NRBC levels above 1100/µL died. In addition, NRBC-positive patients had a significantly lower pO_2_ levels during intensive care than NRBC-negative patients. Low pO_2_ levels appeared to precede the appearance of NRBCs, especially among patients at high risk of mortality [57].

## 6. NRBCs as a Diagnostic Method for Thalassemia

Thalassemia constitutes a group of inherited, autosomal recessive, hematological disorders causing hemolytic anemia because of decreased or nonexistent globin chain synthesis. The globin chain imbalance causes hemolysis and impairs erythropoiesis. Alpha thalassemia is caused by the reduced or nonexistent synthesis of the globin chains of alpha globin, while beta thalassemia is caused by the reduced or nonexistent synthesis of beta globin chains [58]. Several studies have examined whether the NRBC count test is a valuable tool for the prenatal diagnosis of thalassemia and the optimization of transfusion therapy [59,60,61,62].

A study by Shafei et al. reviewed and compared the diagnostic value of two non-invasive diagnostic methods for assessing prenatal thalassemia using cell-free fetal DNA (cff-DNA) and NRBC counts. It was found that the sensitivity and specificity of diagnosing thalassemia via the cff-DNA method were 100% and 84%, respectively, while those using the NRBC method were 100% and 92%. This led to the conclusion that these methods might be used as a screening test, but because of their lower specificity than chorionic villus sampling, they cannot be used as a diagnostic test [59].

Another method for the non-invasive diagnosis of beta-thalassemia was studied by Wei et al. using a single fetal NRBC sample from mother’s blood, which was compared with a genotype extracted from chorionic villi or amniocytes. The accuracy rate of the described method was 87.5%, thus confirming that it could become an optional approach for the non-invasive prenatal diagnosis of beta-thalassemia [60].

A research study by Karakukcu et al. aimed to evaluate the correlation of NRBCs with ineffective erythropoiesis and used it to optimize transfusion therapy among patients with beta thalassemia major. Analysis of the data from this research determined that NRBCs were present in the blood of all the patients with thalassemia major and 87% of patients with thalassemia intermedia; the others had very mild clinical features of the disease and a favorable genetic profile. The data collected suggested that the number of NRBCs may be a useful parameter with respect to the degree of ineffective erythropoiesis [61].

Another study attempted to establish a correlation between NRBCs and ineffective erythropoiesis in various thalassemia syndromes. The number of NRBCs showed a positive correlation with ineffective erythropoiesis: the highest numbers were observed in patients with thalassemia syndromes with almost completely ineffective erythropoiesis. NRBCs were not present among patients affected by hereditary spherocytosis, which is also a disease involving efficient erythropoiesis. In conclusion, NRBC counts may be useful for improving our understanding of ineffective erythropoiesis among patients with thalassemia and may also help to optimize transfusion therapy for patients with severe thalassemia syndromes [62] (Table 1).

## 7. Conclusions

An NRBC count is a basic and inexpensive laboratory test that is simple, cost-effective, and highly reliable. It is currently rarely used in everyday clinical practice, but scientific reports attest to its potential usefulness in the diagnostic process, especially in the context of predicting a patient’s clinical outcome. The most promising reports suggest that in the pediatric population, NRBC counts can be used as an indicator of neonatal hypoxia, the occurrence and consequences of asphyxia, and overall neonatal mortality. In the adult population, NRBCs can be used as a marker for the clinical prognosis of critically ill patients and as a predictor of patient mortality. This is especially relevant for critically ill patients hospitalized in the ICU and with additional risk factors such as the occurrence of ARDS, sepsis, acute pancreatitis, or cardiac complications. More research is required to better investigate the exact role of nucleated red blood cells as an independent prognostic marker. 

## Figures and Tables

**Figure 1 cells-12-01817-f001:**
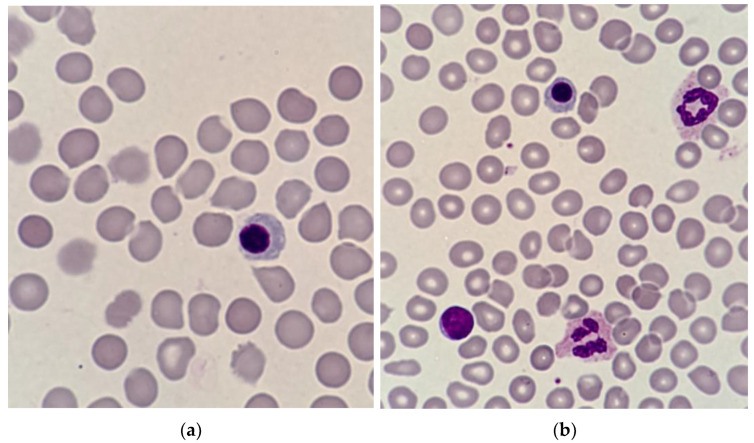
Microscopic images showing immature nucleated red blood cells in the bone marrow. (**a**) An erythroblast surrounded by mature erythrocytes; (**b**) Bone marrow preparation containing an erythroblast, erythrocytes, neutrophils, and a lymphocyte. Source: Alicja Siwicka, PhD, Cytohematology Laboratory, Pediatric Hospital of the Medical University of Warsaw, Poland.

**Figure 2 cells-12-01817-f002:**
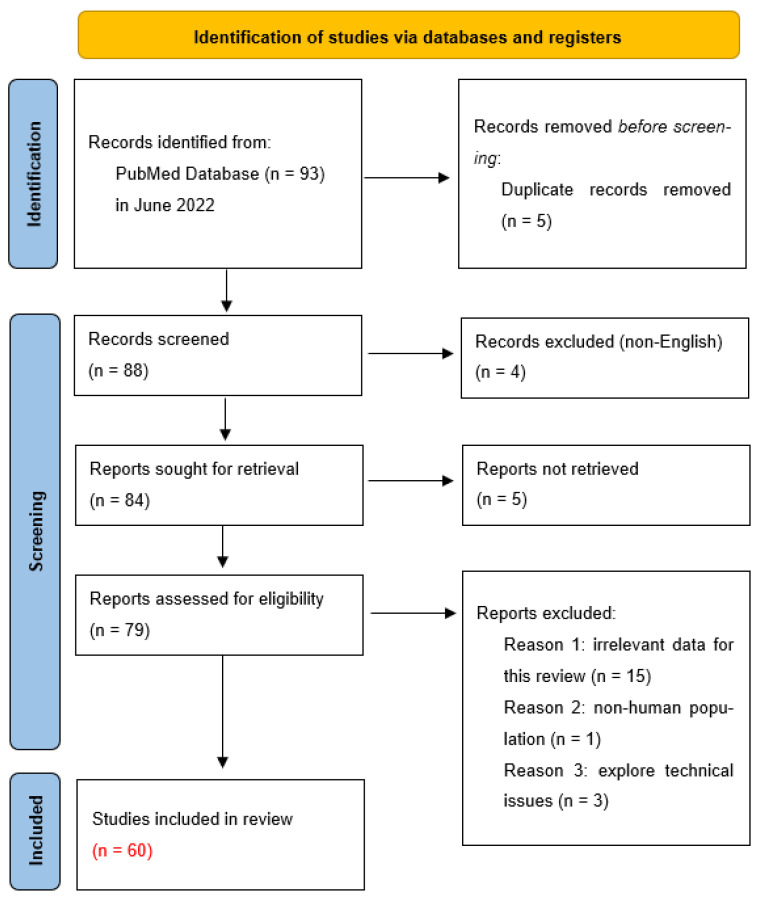
PRISMA flow diagram (2020). Adapted from Page et al. [26].

**Table 1 cells-12-01817-t001:** Summary of disease groups, individual medical entities, and conditions in which an increase in NRBC levels can be observed.

↑NRBC	
Group	Disorders
Neonatal conditions	Hypoxia [16,29,30,31]
	Asphyxia [17,47,48,50]
	Mortality [18,41,43,44,45]
Hematological diseases	Chronic myeloid leukemia [23]
	Myelodysplastic syndromes [51]
	Thalassemia [59,60,61,62]
Critical conditions	Trauma [22,56]
	Sepsis [55]
	Acute pancreatitis [54]
	ARDS [53]
	Hypoxemia [57]
	Hospitalization in the ICU [52]

## Data Availability

No applicable.

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
