# Peer review of "Diagnostic Value and Prognostic Significance of Nucleated Red Blood Cells (NRBCs) in Selected Medical Conditions"

_cells, 2023, doi:10.3390/cells12141817_

Round 1
Reviewer 1 Report
This paper reviews the literature, up to June 2022 in PubMed, on the diagnostic value and prognostic significance of nucleated RBCs in selected medical conditions.
Major comments:
1. It is not true that adult peripheral blood normally contains zero nucleated RBCs. Some immature RBCs are routinely released into the peripheral blood of healthy blood donors. CD34+ cells can be isolated from peripheral blood mononuclear cells in the buffy coat of blood donors and cultured to produce mature RBCs (Griffiths et al. Maturing reticulocytes internalize plasma membrane in glycophorin A-containing vesicles that fuse with autophagosomes before exocytosis. Blood. 2012 Jun 28;119(26):6296-306).
2. It would be helpful to clearly define exactly what the authors mean by NRBCs. Are these CD34+ cells or CD235a+ cells? Define which NRBCs are included in this review.
3. It would also be useful to explain the usual cause of release of NRBCs from the bone marrow in the introduction – i.e. in healthy adult individuals a very small number of NRBCs constantly leak from the bone marrow. The authors already state that “NRBCs can be found in fetal circulation and seem to disappear in healthy neonates in the first month of life [7–9]”. However, low oxygen (or reduced RBC numbers) leads to increased erythropoiesis and increased release of immature RBCs in both adults and neonates.
4. “NRBC count of more than 11 per 100 WBCs or above the 1554 absolute value were shown to have high sensitivity and specificity in predicting 248 complications of asphyxia.” Define the 1554 absolute value.
5. “Concomitant 300 occurrence of negative NRBC test result and the Charlson Complexity Index (CCI) below 7 led to the survival in patients with acute pancreatitis.” Should be “Concomitant 300 occurrence of negative NRBC test result and the Charlson Complexity Index (CCI) below 7 predicted the survival in patients with acute pancreatitis.”
Minor comments:
1. “Exhaustive research was made in June 2022 in the PubMed online electronic database 62 using the key phrases "nucleated red blood cells" or "NRBC".” Would be better as “Extensive research was made in June 2022 in the PubMed online electronic database using the key phrases "nucleated red blood cells" or "NRBC".”
2. Define terms: PRISMA, EEG, APACHE,
3. Typo in Figure 1: “Studies included in review (n = 5)” should be “Studies included in review (n = 52)”.
4. Typos in text: “The study retrospectively analyzed the cases of neonates born form 34 weeks gestation who were given two doses a dose of darbepoetin after birth.” should be “The study retrospectively analyzed the cases of neonates born from 34 weeks gestation who were given two doses or a dose of darbepoetin after birth.”
5. A number of words split by hyphens: signif-icantly, In-creased, demon-strated, oxy-genation, pa-rameter,
6. “Persson et al. studied that also the risk of severe asphyxia in newborns born at term increases with maternal 166 overweight and obesity,” should be “Persson et al. suggested that also the risk of severe asphyxia in newborns born at term increases with maternal overweight and obesity,”
7. The abbreviation “emergency department (ED)” is unnecessary, write in full.
8. Typo: “ARSD” should be “ARDS”.
9. “mcL” and “µmL” are not standard units. Should these both be “µL” or “ml”?
10. Typo: “possitive”.
11. All the references need checking, some are incomplete – e.g. 3, 6, 8, 18, 24, 25, 29, 30, 35, 39, 42, 43, 45, 49, 50, 56, 58. To avoid errors and omissions it is easier to copy and paste the ‘Cite’ reference in PubMed.
Reviewer 2 Report
General
The authors have reviewed an area of clinical haematology that is emerging as one with considerable prognostic value in various clinical states, especially in paediatrics.
Major comments
1. The authors owe it to the reader(s) to: (1) show an image of a series of typical NRBCS, and (2) to show a field of view under a light microscope of NRBCS sitting amongst their white cell neighbours, and RBCS. This will convey to the readers an impression both of the scale of the pathophysiological changes in the blood, and the diagnostic (observational) challenge faced by the laboratory haematologist.
2. I liked/apprecaited the description of the strategy used to review the literature that is encapsulated in Figure 1. This is a fine example of how such reviews should be formulated.
3. Since this is a Review of the literature and the number of papers reviewed is declared it would be useful to somehow convey a checklist…sort of “ticking off” of each paper as it is dealt with. Could a numbering system for the checklist be used, over and above the reference list? This is me thinking out loud as I have not considered this question before. In other words I would like to know that of the 54 papers declared to have been reviewed that each of them has been mentioned in the text somewhere.
Minor criticisms
I have tried to be (painfully for me) thorough. These “trapped” peccadilloes are really too numerous and yet I pressed on knowing that English is likely to not be the first language of the authors. However, many of the points are not specific to English speakers/writers. If the authors learn from this (without wanting to seem patronizing) and the next paper is better prepared, then my efforts on this Sunday morning will have been worth it!
P1 L31 insert ‘the’ before “fetal”
P1 L40 “others” -> ‘other’
P1 L34 “an increased” -> ‘increased’
P1 L43 “revealed” -> ‘reveal’
P1 L43 “exponent” -> ‘indicator’
P2 L56 insert comma after “neonate”
P2 L60 “in such extent in”-> ‘to such an extent in the’
P3 L128 “form” -> ‘from’
P4 L129 “doses a dose” -> ‘doses’ (?)
P4 l133 “seemed to conclude” -> ‘concluded’
P4 L149 remove hyphen
P4 L151 NOTE here and throughout the test the authors are inconsistent in their use of numerals and words for numbers. I recommend the following “Rule” for scientific/medical texts: use the Arabic numeral for any value that has physical units (s, min, h, d, y; mg, g, kg etc) and for any other item like “rats” if the number is less than ten (‘hence 9 rats and ten mice’). AND use the agreed SI units (s, min, h, d, y; mg, g, kg etc)…throughout the text.
Inconsistent nomenclature is annoyingly distracting and reflects a lack of attention to detail which leaves the reader in a state-of-mind that questions even the conclusions drawn by the authors. If they can’t be trusted to attend to trivial presentational details can they be trusted to analyse data consistently?
P4 L151…mixed words and numerals…fix throughout the WHOLE text
P4 L160 & L179 “cesarian” -> ‘Caesarian’ (this is how his name was spelt in Roman times!)
P4 L164 remove the hyphen
P4 L167 “indicates” -> ‘suggests’
P5 L199 “5 days” -> (authors got the first part right but not the second) ‘5 d’
P5 L201 “10/nL” (note here the correct use of units, nL,…but is this use of “nL” typical in hematology Labs now?…I am used to mm^-3 as the basic volume used…please clarify)
P5 L212 “NRBC” -> insert ‘count’ after “NRBC”
P5 L215 “lay” -> ‘lie’
P5 L218 remove hyphen
P5 L227 “evaluation” -> ‘elevation’
P5 L252 “levels” -> ‘counts’ (be consistent in terminology…avoid “level” in science entirely when referring to count-numbers and concentrations etc !)
P5 L253 (as noted above) “hours” -> ‘h’
P5 L261 insert ‘the’ before “English-“
P5 L262 “NRBC” -> ‘NRBCs’; population” -> ‘populations’
P5 L263 “imatinib”->’Imatinib’
P6 L280 “serval” -> ‘several’
P6 L285 use the “Oxford comma” in scientific-article lists (throughout the document) hence a comma before the “and” (Is this the House Style of the Journal?…perhaps it should be?)
P7 L294 “in regards to” -> ‘with regard to’
P7 L301 “Charlson Compexity Index” -> do the authors mean ‘Charlson Comorbidity Index’?
P7 L303 “positive” -> ‘a positive’
P7 L306 “NRBC” -> ‘NRBC count’
P7 L309 & 311 “mcL” I assume means microlitre (not milli centilitre?) in which case use the SI standard Greek-letter mu followed immediately by ‘L’
P7 L314 insert ‘counts’ after “NRBC”
P7 L322 “tension” -> ‘pressure’
P7 L327 “mu-mL” -> now here we see the proper use of Greek-mu but is the “m” intended”. Please make all units consistent and uniform throughout the text. If this article is intended to be used by clinicians, then the units need to be correctly expressed in a logical way throughout the entire text, even if the authors of the original papers did this in a haphazard way. If this review is to add value to the literature it can, as one of its “briefs”, at least systematize units that are used to express the disparate results; otherwise, why have such a review?
P7 L343 insert ‘count’ after “NRBC”
P8 L349 insert ‘sample’ after “NRBC” ? Is this is what was meant?
P8 L359 delete the hyphen in the word
P9 L377 insert ‘count’ after “NRBC”
P9 L379 “NRBCs” -> ‘NRBC count’
P9 L382 (since we are being pedantic!...split infinitive) “to better investigate” -> ‘to investigate better’
Overall
This is a timely review, but it needs better contextualizing of the subject matter. In other words, it needs pictures/images of nucleated RBCs and better explanations of the diagnostic methodology. And, it needs to be better proofread (next time…I have done it for the authors this time; but this is getting tiresome for me!...ChatGPT might well do it next time…or the Journal Office might consider editing for typos etc before sending such manuscripts out for review?)
General
The authors have reviewed an area of clinical haematology that is emerging as one with considerable prognostic value in various clinical states, especially in paediatrics.
Major comments
1. The authors owe it to the reader(s) to: (1) show an image of a series of typical NRBCS, and (2) to show a field of view under a light microscope of NRBCS sitting amongst their white cell neighbours, and RBCS. This will convey to the readers an impression both of the scale of the pathophysiological changes in the blood, and the diagnostic (observational) challenge faced by the laboratory haematologist.
2. I liked/apprecaited the description of the strategy used to review the literature that is encapsulated in Figure 1. This is a fine example of how such reviews should be formulated.
3. Since this is a Review of the literature and the number of papers reviewed is declared it would be useful to somehow convey a checklist…sort of “ticking off” of each paper as it is dealt with. Could a numbering system for the checklist be used, over and above the reference list? This is me thinking out loud as I have not considered this question before. In other words I would like to know that of the 54 papers declared to have been reviewed that each of them has been mentioned in the text somewhere.
Minor criticisms
I have tried to be (painfully for me) thorough. These “trapped” peccadilloes are really too numerous and yet I pressed on knowing that English is likely to not be the first language of the authors. However, many of the points are not specific to English speakers/writers. If the authors learn from this (without wanting to seem patronizing) and the next paper is better prepared, then my efforts on this Sunday morning will have been worth it!
P1 L31 insert ‘the’ before “fetal”
P1 L40 “others” -> ‘other’
P1 L34 “an increased” -> ‘increased’
P1 L43 “revealed” -> ‘reveal’
P1 L43 “exponent” -> ‘indicator’
P2 L56 insert comma after “neonate”
P2 L60 “in such extent in”-> ‘to such an extent in the’
P3 L128 “form” -> ‘from’
P4 L129 “doses a dose” -> ‘doses’ (?)
P4 l133 “seemed to conclude” -> ‘concluded’
P4 L149 remove hyphen
P4 L151 NOTE here and throughout the test the authors are inconsistent in their use of numerals and words for numbers. I recommend the following “Rule” for scientific/medical texts: use the Arabic numeral for any value that has physical units (s, min, h, d, y; mg, g, kg etc) and for any other item like “rats” if the number is less than ten (‘hence 9 rats and ten mice’). AND use the agreed SI units (s, min, h, d, y; mg, g, kg etc)…throughout the text.
Inconsistent nomenclature is annoyingly distracting and reflects a lack of attention to detail which leaves the reader in a state-of-mind that questions even the conclusions drawn by the authors. If they can’t be trusted to attend to trivial presentational details can they be trusted to analyse data consistently?
P4 L151…mixed words and numerals…fix throughout the WHOLE text
P4 L160 & L179 “cesarian” -> ‘Caesarian’ (this is how his name was spelt in Roman times!)
P4 L164 remove the hyphen
P4 L167 “indicates” -> ‘suggests’
P5 L199 “5 days” -> (authors got the first part right but not the second) ‘5 d’
P5 L201 “10/nL” (note here the correct use of units, nL,…but is this use of “nL” typical in hematology Labs now?…I am used to mm^-3 as the basic volume used…please clarify)
P5 L212 “NRBC” -> insert ‘count’ after “NRBC”
P5 L215 “lay” -> ‘lie’
P5 L218 remove hyphen
P5 L227 “evaluation” -> ‘elevation’
P5 L252 “levels” -> ‘counts’ (be consistent in terminology…avoid “level” in science entirely when referring to count-numbers and concentrations etc !)
P5 L253 (as noted above) “hours” -> ‘h’
P5 L261 insert ‘the’ before “English-“
P5 L262 “NRBC” -> ‘NRBCs’; population” -> ‘populations’
P5 L263 “imatinib”->’Imatinib’
P6 L280 “serval” -> ‘several’
P6 L285 use the “Oxford comma” in scientific-article lists (throughout the document) hence a comma before the “and” (Is this the House Style of the Journal?…perhaps it should be?)
P7 L294 “in regards to” -> ‘with regard to’
P7 L301 “Charlson Compexity Index” -> do the authors mean ‘Charlson Comorbidity Index’?
P7 L303 “positive” -> ‘a positive’
P7 L306 “NRBC” -> ‘NRBC count’
P7 L309 & 311 “mcL” I assume means microlitre (not milli centilitre?) in which case use the SI standard Greek-letter mu followed immediately by ‘L’
P7 L314 insert ‘counts’ after “NRBC”
P7 L322 “tension” -> ‘pressure’
P7 L327 “mu-mL” -> now here we see the proper use of Greek-mu but is the “m” intended”. Please make all units consistent and uniform throughout the text. If this article is intended to be used by clinicians, then the units need to be correctly expressed in a logical way throughout the entire text, even if the authors of the original papers did this in a haphazard way. If this review is to add value to the literature it can, as one of its “briefs”, at least systematize units that are used to express the disparate results; otherwise, why have such a review?
P7 L343 insert ‘count’ after “NRBC”
P8 L349 insert ‘sample’ after “NRBC” ? Is this is what was meant?
P8 L359 delete the hyphen in the word
P9 L377 insert ‘count’ after “NRBC”
P9 L379 “NRBCs” -> ‘NRBC count’
P9 L382 (since we are being pedantic!...split infinitive) “to better investigate” -> ‘to investigate better’
Overall
This is a timely review, but it needs better contextualizing of the subject matter. In other words, it needs pictures/images of nucleated RBCs and better explanations of the diagnostic methodology. And, it needs to be better proofread (next time…I have done it for the authors this time; but this is getting tiresome for me!...ChatGPT might well do it next time…or the Journal Office might consider editing for typos etc before sending such manuscripts out for review?)
Reviewer 3 Report
In this manuscript, Pikora and colleagues review the prognostic value of nucleated red cells in various neonatal and adult conditions. It is generally well written and likely to be of interest to readers in several fields.
The manuscript may be improved by addressing the following concerns:
1. The title is a bit misleading as nucleated red cells do not appear to be diagnostic for any given condition.
2. A paragraph describing how the nucleated red cell count is currently integrated into clinical decision making would be useful.
3. A table listing the conditions in which the nucleated red cell count is prognostic, along with the additional tests required to compensate for its lack of specificity in each condition, would be helpful. For example, how does one decide in the neonatal period whether an elevated nucleated red cell count indicated hypoxia or asphyxia? To what degree does either of these contribute to its usefulness as a biomarker of mortality?
4. Figure 1 states that 5 studies were included in the review. Presumably this needs to be corrected.
No comments
Author Response
Please see the attachement.
